# Crosslinked Sulfonated Polyphenylsulfone-Vinylon (CSPPSU-vinylon) Membranes for PEM Fuel Cells from SPPSU and Polyvinyl Alcohol (PVA)

**DOI:** 10.3390/polym12061354

**Published:** 2020-06-16

**Authors:** Je-Deok Kim, Satoshi Matsushita, Kenji Tamura

**Affiliations:** 1Polymer Electrolyte Fuel Cell Group, Global Research Center for Environmental and Energy Based on Nanomaterials Science (GREEN),Tsukuba Ibaraki 305-0044, Japan; satoshi.matsushita@agc.com; 2Hydrogen Production Materials Group, Center for Green Research on Energy and Environmental Materials, Tsukuba Ibaraki 305-0044, Japan; 3Functional Clay Materials Group, Research Center for Functional Materials, National Institute for Materials Science (NIMS), 1-1 Namiki, Tsukuba, Ibaraki 305-0044, Japan; Tamura.Kenji@nims.go.jp

**Keywords:** PPSU, SPPSU, PVA, CSPPSU-vinylon, PEMFCs

## Abstract

A crosslinked sulfonated polyphenylsulfone (CSPPSU) polymer and polyvinyl alcohol (PVA) were thermally crosslinked; then, a CSPPSU-vinylon membrane was synthesized using a formalization reaction. Its use as an electrolyte membrane for fuel cells was investigated. PVA was synthesized from polyvinyl acetate (PVAc), using a saponification reaction. The CSPPSU-vinylon membrane was synthesized by the addition of PVA (5 wt%, 10 wt%, 20 wt%), and its chemical, mechanical, conductivity, and fuel cell properties were studied. The conductivity of the CSPPSU-10vinylon membrane is higher than that of the CSPPSU membrane, and a conductivity of 66 mS/cm was obtained at 120 °C and 90% RH (relative humidity). From a fuel cell evaluation at 80 °C, the CSPPSU-10vinylon membrane has a higher current density than CSPPSU and Nafion212 membranes, in both high (100% RH) and low humidification (60% RH). By using a CSPPSU-vinylon membrane instead of a CSPPSU membrane, the conductivity and fuel cell performance improved.

## 1. Introduction

In order to realize a low-carbon society that includes highly efficient energy systems which make effective use of renewable energy, economically sustainable growth, environmental protection, and energy security are required. Energy conversion/storage devices—such as fuel cells, water electrolysis, secondary batteries, and solar cells—are core technologies for building a low-carbon society. In order to produce these devices safely and with high performances, their constituent materials must have high performances. Polymer electrolyte membranes for proton exchange in fuel cells are required for polymer electrolyte membrane fuel cells (PEMFCs), and polymer electrolyte membrane water electrolysis (PEMWE), and Nafion membranes mainly using perfluorosulfonic acid (PFSA) ion exchange resins, are often used. Nafion membranes have high proton conductivities and excellent chemical stabilities, and have been used in both mobile and stationary fuel cells [1,2]. However, the performance of these fuel cells suffers from the deterioration of their mechanical properties, due to thinning of the electrolyte membrane. In addition, higher operating temperatures are required to improve both the glass transition temperature (*T*_g_) and the proton conductivity at high temperatures. At the same time, research on hydrocarbon-based electrolytes, instead of fluorine-based electrolytes, has been conducted. Non-fluorine-based electrolytes are low cost materials, have high *T*_g_ values, and have been studied for many years. However, their performances and chemical stabilities, which are lower than those of fluorine-based ones, have been the biggest obstacles towards their practical use. Thus, materials for proton exchange electrolytes with higher performances are still needed.

The drawbacks of PFSA membranes have prompted research into alternative membranes. Various aromatic polymer ionomer membranes are being actively investigated. Sulfonated polyphenylsulfone (SPPSU) [3,4,5,6,7,8,9,10,11,12,13,14,15,16,17,18,19,20], sulfonated polyetheretherketone (SPEEK) [21,22,23,24,25,26,27,28,29,30,31,32], sulfonated polysulfone (SPSU) [33,34,35,36,37,38], sulfonated polyphenylene sulfone (SPPS) [39], sulfonated polyphenylene (SPP) [40], sulfonated polyethersulfone (SPES) [41,42], sulfonated polyimide (SPI) [43,44,45], sulfonated polyphenylene oxide (SPPO) [46], and polybenzimidazole (PBI) [47,48] are attracting special interest. 

We are developing a crosslinked sulfonated polyphenylsulfone (CSPPSU) membrane using the sulfonation of the polyphenylsulfone (PPSU) polymer, which has an excellent thermal stability, high chemical resistance, and is low cost [3,5,6,19,20]. According to a fuel cell evaluation using the CSPPSU membrane, the membrane could be used for 4000 h [19]. However, compared to fluorine-based electrolyte membranes, hydrocarbon-based CSPPSU membranes are still insufficient in terms of high performances and high durabilities in fuel cells. It is necessary to further improve the performance and durability of SPPSU membranes. In this study, we prepared a CSPPSU-vinylon membrane, which has a higher performance than both CSPPSU and Nafion212 membranes. Vinylon was obtained using a formalization reaction with polyvinyl alcohol (PVA), and PVA was obtained by the saponification of polyvinyl acetate (PVAc). Crosslinked SPPSU-vinylon membranes from the SPPSU and PVA, were obtained using thermal crosslinking and a formalization reaction. The vinylon was somewhat stable, even in a high-hydration environment. First, we prepared a crosslinked SPPSU-vinylon membrane using thermal crosslinking and the vinylonization of a SPPSU-PVA composite, and these methods appeared promising for the thinning of other polymer electrolyte membranes.

## 2. Experimental

### 2.1. Materials

Polyvinyl acetate (PVAc, (C_4_H_6_O_2_)_n_, Mw = 100,000) was purchased from Sigma-Aldrich Corporation (St. Louis, MO, USA). A DuPontTM Nafion212 membrane (NR-212) was purchased from DuPont (USA). Methanol (CH_3_OH), formaldehyde (CH_2_O, 37%), sodium chloride (NaCl), sodium hydroxide (NaOH), and sulfuric acid (H_2_SO_4_) were purchased from Nacalai Tesque, Inc. Polyphenylsulfone (Solvay Radel R-5000 NT) (Mn = 26,000; Mw = 50,000; Mw/Mn = 1.9) was provided by Solvay Specialty Polymers Japan K.K. (glass transition temperature (T_g_) = 220 °C, (Tokyo, Japan). A dialysis tubing cellulose membrane, which has a molecular weight cut-off (MWCO) of 14,000, and dimethyl sulfoxide (DMSO) were purchased from Sigma-Aldrich Co., Ltd. Deionized (DI) water was obtained using a PURELAB^®^ Option-R 7 ELGA LabWater, at 15 Mohm cm and 25 °C. Sodium sulfate (Na_2_SO_4_) and Iron (II) chloride tetrahydrate (FeCl_2_·4H_2_O) were purchased from Fujifilm Wako Pure Chemical Corporation (Osaka, Japan). 

### 2.2. Synthesis of SPPSU and PVA, and Preparation of CSPPSU-vinylon Membranes

The synthesis and properties of SPPSU (Mw ≈ 150,000) have been described in detail in previous reports [3,19]. PVA was synthesized using the following method. PVAc (1 g) was dissolved in a flask with methanol (50 mL). Then, a 40% NaOH solution was added, and the mixture was allowed to react at 40 °C for 10 min (saponification). The reaction mixture was washed with methanol 4 times and filtered. To remove the remaining solvent, the product was dried at 80 °C for 24 h, and PVA (0.5 g, Mw ≈ 50,000) was obtained. Crosslinked SPPSU-vinylon membranes were obtained from SPPSU and PVA using the following method. A glass vial was charged with SPPSU (0.5 g) and DMSO (20 mL), and dissolved. PVA (0.025 g, 0.05 g, 0.1 g) and DMSO (4 mL) were put into another glass vial and dissolved. Then, the PVA-DMSO solution was added to the SPPSU-DMSO solution, and the mixture was stirred for 1 h. The SPPSU-PVA-DMSO solution was transferred to a glass container, dried for 24 h at 80 °C, and then annealed in air at 120 °C (24 h), 160 °C (24 h), and 180 °C (24 h). Next, the vinylon from the PVA was prepared using a formalization solution (H_2_O:H_2_SO_4_:Na_2_SO_4_:CH_2_O = 1.00:0.21:0.20:0.06 in mass ratio) reaction, for 2 h at 60 °C. Activation was performed using the following procedure: heating in 0.5 M NaOH at 80 °C overnight, washing in DI H_2_O, heating at 1M H_2_SO_4_ at 80 °C for 2 h, and washing in DI H_2_O for 2 h. Finally, the crosslinked SPPSU-vinylon membranes were dried at room temperature before use. The crosslinked SPPSU-vinylon membranes were very flexible and dark brown. The classification of the crosslinked SPPSU-vinylon membranes is shown in Table 1.

### 2.3. Iron-Exchange Chromotography (IEC), D.S. (Degree of Sulfonation), Water-Uptake (W.U.), λ, and Crosslink Rates (D_crosslink_)

The IEC values were determined using the following equation: IEC (meq/g) = c*v*/W_dry_, where c (mmol/L) is the concentration of standardized NaOH aq. used for titration (0.01mol/L), *v* (L) is the volume of standardized NaOH aq. used for titration, and W_dry_ (g) is the mass of the dry membrane. The water-uptake (W.U.) of the membranes at room temperature was calculated using the following: W.U. (%) = [(W_wet_ – W_dry_)/W_dry_] × 100, where W_wet_ is the mass of the wet membrane. The hydration number (λ) for the membranes was determined using the following: λ ([H_2_O]/[SO_3_H]) = [1000(W_wet_ – W_dry_)]/18W_dry_IEC. The degree of crosslinking (crosslink rate, D_crosslink_) in the membranes was determined using the following: D_crosslink_ (%) = [(IEC_before annealing_ – IEC_after annealing_)/IEC_before annealing_] × 100 [19].

### 2.4. Oxidative Stability (Fenton’s Test)

The oxidative stabilities of the membranes were evaluated by immersing a small piece of the membrane into Fenton’s reagent [3 wt% H_2_O_2_ and 2 ppm Fe(II) (added as FeCl_2_·4H_2_O)], at ~80 °C for 1 h while stirring. The samples were dried at 80 °C before the measurements. The membranes were repeatedly washed with DI H_2_O, and dried at 80 °C overnight following the reaction. The oxidative stabilities were determined as follows: [(mass of residual membrane after the test)/(initial mass of membrane)] × 100.

### 2.5. Chemical Structure of the Samples

Fourier-transform infrared (FTIR) absorption spectra of the samples were obtained on a Thermo Scientific Nicolet 6700 spectrometer, in an attenuated total reflection (ATR) mode.

### 2.6. Mechanical and Thermal Behavior of the Samples

Stress-strain tests on the membranes were accomplished using a Tension Test Machine (Shimazu, EZ-S) at room temperature [19]. The thermal and mass properties of the membranes were investigated using thermogravimetric and mass analyses with a Thermoplus TG8120 TG-DTA/H (Rigaku Co. Ltd., Japan). The samples were heated from 60 to 800 °C at 5 °C/min in air, after keeping them for 1 h at 60 °C.

### 2.7. Conductivity and Single Cell Measurements of the Samples

The proton conductivities of the membranes were evaluated using a four-point probe impedance spectroscopy. For the membrane electrode assembly (MEA), the thickness of the membranes was approximately 50 μm, and a Pt/C/ionomer (ionomer/carbon = 1) catalyst electrode (EIWA corporation) containing 0.3 mg/cm^2^ of Pt on a GDL electrode (Sigracet^®^ GDL 25BC of SGL Group Co. Ltd., Japan), was used. The effective electrode area of the single cell was 4 cm^2^. The MEA was acquired by loading a membrane between the anode and cathode, and hot-pressing at 130 °C and ~9.8 kN for 20 min. A single cell performance was measured in relation to the amount of hydrogen (H_2_) and oxygen (O_2_) at the anode and cathode, respectively, at 80 °C, 100% and 60% RH (relative humidity), and ambient pressure. The gas flow rate of hydrogen and oxygen were 50 cc/min and 100 cc/min, respectively. Linear sweep voltammetry (LSV) was evaluated in the potential range of 0.02–0.5 V at 2 mV/s [19]. 

## 3. Results and Discussion

### 3.1. CSPPSU-vinylon Membranes

Thermally crosslinked membranes of SPPSU polymers have been reported in previous research [5,19]. Crosslinking occurs between the sulfone groups of SPPSU under a thermal environment. The same phenomenon occurs in composite membranes of SPPSU polymers and PVA polymers. In addition, crosslinking occurs between the sulfone groups of SPPSU and the hydroxy groups of PVA upon heat treatment. A crosslinked SPPSU-vinylon membrane could be obtained using a formalization reaction with PVA (Figure 1).

PVA was synthesized by hydrolyzing PVAc (saponification). The peak for the carbonyl groups (C=O) of the PVAc appeared at 1728 cm^−1^ in the IR spectrum (Figure 2a). For PVA synthesized using the hydrolysis of PVAc, the peak due to the carbonyl group disappeared, and new peaks for the OH groups appeared at 3272 cm^−1^, 1655 cm^−1^, and 1324 cm^−1^ (Figure 2b). These results indicate that PVA can be obtained by hydrolyzing PVAc. The IR spectra of the CSPPSU-vinylon membranes did not change significantly with the amount of PVA added and had similar characteristics (Figure 2c–e). The peaks for both SPPSU and PVA appeared in the spectra. The crosslinking of SPPSU has been reported in more detail in a previous paper [3,5]. In the IR spectra, it was not clear whether the sulfone bridge of SPPSU and the hydroxy group of PVA were crosslinked using hydrolysis to form a sulfone bridge (-SO_2_-). Moreover, it was difficult to determine whether PVA had been changed to vinylon. However, we can assume that crosslinking and vinylon formation progressed, as the appearance of the obtained membranes were very uniform and flexible. Figure 2 and Table 2 show the FTIR spectra and summarize the assignments of the peaks, respectively.

### 3.2. Thermal and Mechanical Properties of the CSPPSU-vinylon Membranes

The thermal (Figure 3) and mechanical properties (Figure 4) of the CSPPSU-vinylon membranes, prepared by varying the amount of PVA added, were investigated. The CSPPSU-vinylon membrane exhibited a lower desorption of the sulfone groups and had a lower decomposition temperature of the polymer backbone than the CSPPSU membrane (Figure 3, Table 3). The thermal behavior of the CSPPSU-vinylon membrane on the amount of PVA added was similar. As for the weight reduction ratio of water due to water vaporization, the CSPPSU-vinylon membrane had a higher water content than the CSPPSU membrane (Table 3). In the TGA (thermal gravimetric analysis) curves for the CSPPSU sample, residuals (inorganic substances) appeared after 600 °C. However, every CSPPSU-vinylon sample burned at 600 °C. This suggests that the SPPSU and the vinylon were crosslinked into one polymer.

On the other hand, the dependence of the CSPPSU-vinylon membrane on the amount of added PVA, was noticeable in the evaluation of its mechanical properties (Figure 4, Table 4). The CSPPSU-10vinylon membranes obtained by adding 10 wt% PVA to SPPSU had higher tensile strengths than the other crosslinked membranes. However, the tensile elongation increased with an increase in the amount of added PVA, and the tensile strength and tensile elongation of the CSPPSU-5vinylon membrane containing 5 wt% PVA, was low in comparison to the other membranes. The flexural modulus of the membrane decreased with an increase in the amount of PVA. The tensile elongation characteristics of the CSPPSU-vinylon membrane were smaller than those of the CSPPSU membranes. The favorable tensile elongation of the CSPPSU-10vinylon membrane may be due to its better homogeneity in comparison to the other membranes.

### 3.3. Proton Conductivities of the CSPPSU-vinylon Membranes

Polymer electrolyte membranes for high-performance fuel cells require high proton conductivities of >0.01 S/cm, from low to high temperatures and high to low humidification [1]. The conductivity of the CSPPSU-vinylon membranes due to the difference in the amount of PVA added—which is the average value of the error bars of the data obtained after three measurements with varying humidity, at cell temperatures of 40 and 120 °C—is shown in Figure 5. Table 5 shows the physicochemical properties of the CSPPSU-vinylon membranes, depending on the amount of PVA added. The IEC value of the SPPSU polymer was 3.8 meq/g, and the IEC value of the SPPSU-PVA composite polymer was assumed to be equivalent to the IEC value of the SPPSU polymer. Then, the crosslinking degree (D_crosslink_) of the CSPPSU-vinylon membranes was calculated. Moreover, the chemical stability of the membrane was determined using Fenton’s reagent (3 wt% H_2_O_2_ + 2 ppm Fe (II), 80 °C, 1 h). Since vinylon has a high chemical resistance, we thought that the chemical stability of CSPPSU [19] would be improved by incorporating vinylon. However, as shown in Table 5, there was little improvement. With Fenton’s reagent, the CSPPSU membrane was radically attacked from the edge, but the CSPPSU-10vinylon membrane was attacked from the inside of the membrane, generating a hole. It is thought that the SPPSU part is selectively vulnerable to attack.

On the other hand, the conductivity of the CSPPSU-vinylon electrolyte membrane increased with increases in the temperature and humidity. The conductivity of the CSPPSU-vinylon membrane was higher than that of the CSPPSU membrane. In particular, the CSPPSU-10vinylon membrane had a higher conductivity than the other membranes. The diffusion of protons in the electrolyte membrane depended on the concentration and proton mobility of the sulfonic acid groups (-SO_3_H) in the electrolyte membrane, and became faster as the temperature and humidity were increased. In addition, the nanostructure (conduction path) of the electrolyte membrane was greatly affected. The homogeneity of the CSPPSU-10vinylon membrane was better than that of the other membranes (Figure 4). As shown in Table 5, changes in the IEC values of the CSPPSU-vinylon membrane due to the difference in the amount of PVA added, was small and slightly higher than those of the CSPPSU membrane. However, the water content and the number of water molecules per sulfonic acid group (λ) of the CSPPSU-10vinylon membrane, were higher than those of the other crosslinked membranes. These differences contributed to the high proton conductivity of the CSPPSU-10vinylon membrane. The degree of crosslinking of the CSPPSU-vinylon membrane was 42%–45%. It is possible that the hydroxy groups (-OH) of vinylon in the CSPPSU-vinylon membrane contributed to the proton transfer. From the above, it is clear that the conduction mechanism of the SPPSU-vinylon composite membrane is very complicated.

### 3.4. Fuel Cell Properties using CSPPSU-vinylon Membranes

The performance of fuel cells depends not only on the ionic conductivity of the electrolyte membrane, but also on the interfaces between the electrode layers (catalyst, carbon, ionomer) and between the membrane and the electrode layer [45,49]. Here, the electrode layer and the MEA (membrane electrode assembly) were placed under the same conditions, and only the electrolyte membrane was different. In addition, the measurement conditions for obtaining the current-voltage (I-V) characteristics were the same. The I-V characteristics were evaluated at a cell temperature of 80 °C, and humidities of 100% RH and 60% RH. Figure 6 shows I-V_ir free_ and I-iR loss characteristics, evaluated using CSPPSU-10vinylon, CSPPSU, and Nafion212 membranes. The resistance of the single cell using the CSPPSU-10vinylon membrane was higher than that of the Nafion212 membrane, and lower than that of the CSPPSU membrane (Table 6). This tendency in the level of conductivity is the same as that using only the electrolyte membrane (Figure 5). Moreover, the I-V_iR free_ characteristics showed the same tendency as the resistance characteristics of the unit cell. On the other hand, when using the CSPPSU-10vinylon membrane, a current of 1.5 A/cm^2^ or more was obtained. In the case of Nafion212 and the CSPPSU membranes, the current was less than 1.5 A/cm^2^ at 100% RH, and less than 1 A/cm^2^ at 60% RH. Under high humidification conditions, when a high current was applied, flooding occurred on the cathode side, and the voltage tended to drop sharply. Moreover, under low humidification conditions, the membrane resistance increased due to the drying of the membrane, making it difficult to obtain a high current. However, when the CSPPSU-10vinylon membrane was used, a high current without a sharp drop in voltage was obtained, under both high and low humidity conditions. The CSPPSU-vinylon membrane, therefore, seems to have an excellent water treatment ability. These results suggest that the CSPPSU-vinylon membrane would be suitable for thin membrane applications.

Figure 7 shows the hydrogen crossover characteristics of the CSPPSU-10vinylon, CSPPSU, and Nafion212 membranes. The crossover properties of the CSPPSU-10vinylon membrane are five times lower than those of the Nafion212 membrane, and three times higher than those of the CSPPSU membrane (Table 6). We can assume that the crosslinking of SPPSU with vinylon increases the conduction paths (volume fraction) in the nanophase, and hydrogen crossover is higher than that of the CSPPSU membrane. 

## 4. Conclusions

We focused on improving the performance of CSPPSU membranes with hydrocarbon-based SPPSU polymers, as an alternative electrolyte to fluoropolymer electrolytes. To improve the conductivity and I-V performance properties of CSPPSU membranes, SPPSU and PVA were crosslinked, and CSPPSU-vinylon membranes were synthesized by the formalization of PVA, and compared with Nafion212 and CSPPSU membranes. The conductivities of the CSPPSU-vinylon membranes were higher than those of the CSPPSU membrane. From the results of the fuel cell evaluation, higher current densities than those of Nafion212 and CSPPSU membranes were obtained under both high and low humidification conditions. This is due to the effects of vinylon, and it is thought that the CSPPSU-vinylon membrane has excellent water retention under low humidification conditions. Furthermore, the hydrogen gas crossover properties are lower than those of Nafion212. In other words, the CSPPSU-vinylon membrane would be useful for thin membrane applications.

## Figures and Tables

**Figure 1 polymers-12-01354-f001:**
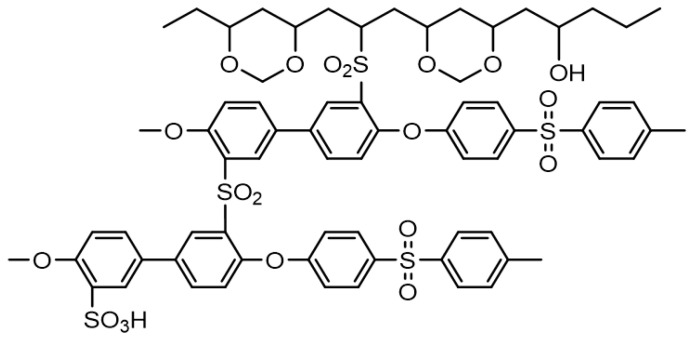
Schematic diagram of CSPPSU-vinylon membrane.

**Figure 2 polymers-12-01354-f002:**
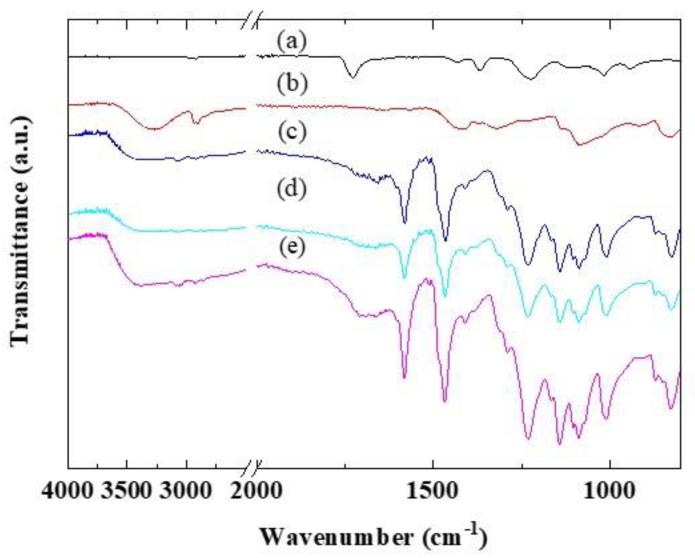
Fourier transform infrared (FTIR) properties of (**a**) polyvinyl acetate (PVAc), (**b**) syn. polyvinyl alcohol (PVA), (**c**) CSPPSU-5vinylon, (**d**) CSPPSU-10vinylon, and (**e**) CSPPSU-20vinylon membranes with different amounts of PVA.

**Figure 3 polymers-12-01354-f003:**
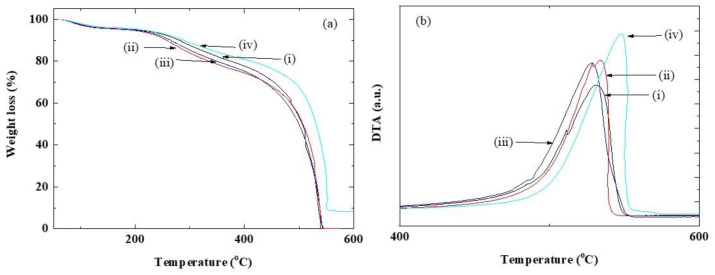
(**a**) TG (thermal gravimetric) and (**b**) DTA (differential thermal analysis) results of (i) CSPPSU-5vinylon, (ii) CSPPSU-10vinylon, (iii) CSPPSU-20vinylon, and (iv) CSPPSU membranes.

**Figure 4 polymers-12-01354-f004:**
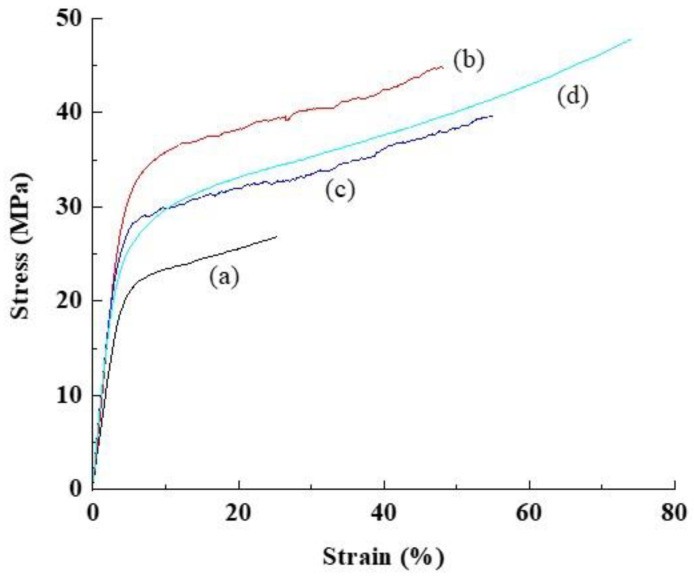
Stress-strain properties of (**a**) CSPPSU-5vinylon, (**b**) CSPPSU-10vinylon, (**c**) CSPPSU-20vinylon, and (**d**) CSPPSU membranes.

**Figure 5 polymers-12-01354-f005:**
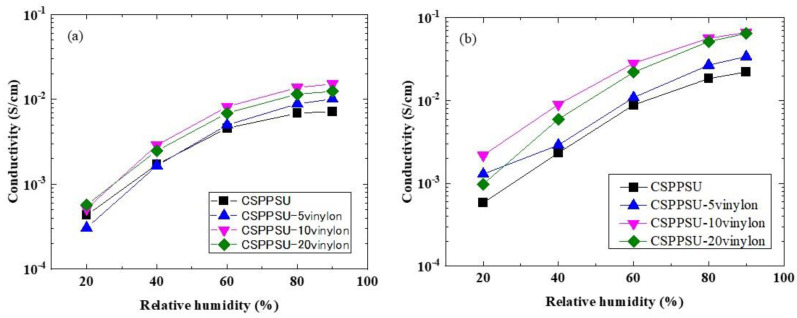
Proton conductivities of the CSPPSU-vinylon membranes vs. the relative humidity at (**a**) 40 and (**b**) 120 °C.

**Figure 6 polymers-12-01354-f006:**
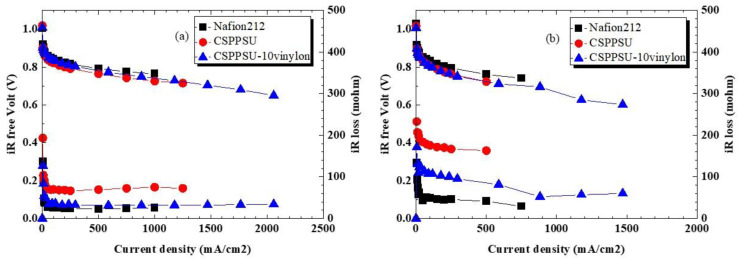
Current-voltage (I-V) properties of CSPPSU-10vinylon, CSPPSU, and Nafion212 membranes: (**a**) I-V_iR free_ at 80 °C, 100% RH and (**b**) I-V_iR free_ at 80 °C, 60% RH.

**Figure 7 polymers-12-01354-f007:**
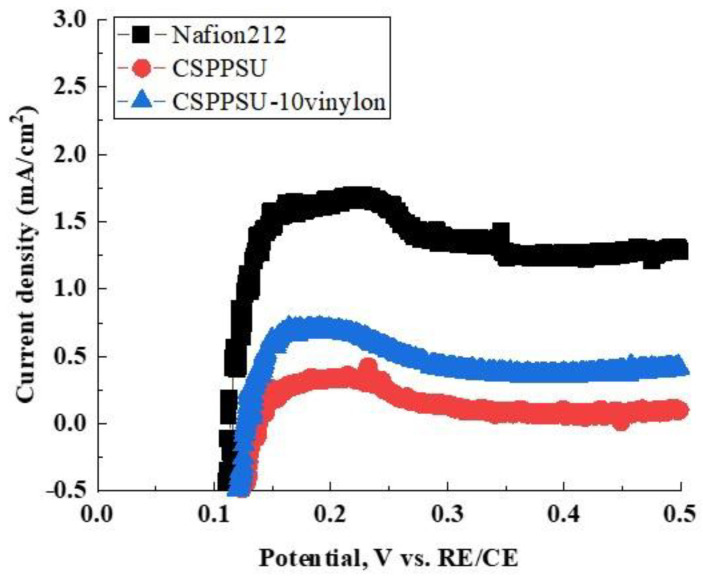
Hydrogen crossover properties of CSPPSU-10vinylon, CSPPSU, and Nafion212 membranes at 80 °C and 100% RH.

**Table 1 polymers-12-01354-t001:** Classification of crosslinked sulfonated polyphenylsulfone-vinylon (CSPPSU-vinylon) membranes.

Varied Parameter	Variable Parameter	Membrane Classification
PVA loading (wt%)	0	CSPPSU
	5	CSPPSU-5vinylon
	10	CSPPSU-10vinylon
	20	CSPPSU-20vinylon

**Table 2 polymers-12-01354-t002:** Summary of assignments of the FTIR spectra of PVAc, syn. PVA, and CSPPSU-vinylon membranes.

Polymer	PVAc	PVA	CSPPSU-vinylon
*v*H-O-H (cm^−1^)		3272	3421
Aromatic *v*C-H (cm^−1^)			3093, 3074, 3038
Aliphatic *v*C-H (cm^−1^)	2974, 2928, 2852, 1431, 1367	2932, 2898, 1424	2920, 2846
δs, H-O-H (cm^−1^)		1655	1714, 1665
Aromatic *v*C=C (cm^−1^)			1584, 1469
*v*C=O (cm^−1^)	1728		
δs, C-O-H (cm^−1^)		1324	1324
*v*as, C-O-C (cm^−1^)			1232
Aliphatic *v*C-O (cm^−1^)	1220, 1116, 1016, 939	1568, 1083	
*v*s, O=S=O (cm^−1^)			1142

**Table 3 polymers-12-01354-t003:** Summary of temperature ranges and mass losses observed for each step in the TG-DTA curves for CSPPSU-vinylon membranes.

Sample Name	Evaporation of H_2_O Interacting with –SO_3_H or -OH Group		Desubstitution of –SO_3_H Group		Thermal Decomposition of Polymer Backbone
	ΔT (°C)	ΔWt. Loss (%)	ΔT (°C)	ΔWt. Loss (%)	Peak of Exothermic (°C)
CSPPSU	61–210	4.5	210–453	19.1	548
CSPPSU-5vinylon	61–212	4.9	212–403	17.5	532
CSPPSU-10vinylon	61–197	5.0	197–389	19.5	534
CSPPSU-20vinylon	61–197	5.0	197–400	19.8	528

**Table 4 polymers-12-01354-t004:** Mechanical properties of CSPPSU-vinylon membranes.

	CSPPSU	CSPPSU -5vinylon	CSPPSU -10vinylon	CSPPSU -20vinylon
Tensile strength (MPa)	48	27	45	40
Tensile elongation (%) (break)	74	26	48	55
Flexural modulus (MPa) *	757	781	759	548

* Δstress/Δstrain.

**Table 5 polymers-12-01354-t005:** Physicochemical and conductivity properties of the CSPPSU-vinylon membranes.

Sample Name			CSPPSU	CSPPSU -5vinylon	CSPPSU -10vinylon	CSPPSU -20vinylon
IEC (meq/g)			2	2.2	2.1	2.1
W.U. (%)			43	38	66	36
λ			11.9	9.6	17.5	9.5
D_crosslink_ (%)			47.3	42.1	44.7	44.7
R_oxidation_ (%)			91–99	_a	81–99	_a
Conductivity (mS/cm)	40 °C	20% RH	0.43	0.31	0.52	0.57
		90% RH	7.23	10.12	15.23	12.43
	80 °C	20% RH	0.7	0.7	0.9	1.1
		90% RH	18	36	56	55
	120 °C	20% RH	0.6	1.3	2.2	1.0
		90% RH	22	34	66	65

_a: disassembly.

**Table 6 polymers-12-01354-t006:** I-V and H_2_ crossover data for single cells using the CSPPSU-10vinylon, CSPPSU, and Nafion212 membranes.

	80 °C, 100% RH			80 °C, 60% RH	
	OCV (V)	iR Loss (mohm) @ 1 A/cm^2^	H_2_ Crossover (mA/cm^2^) @ 0.4 V	OCV (V)	iR Loss (mohm) @ 1 A/cm^2^
CSPPSU	1.020	73	0.085	1.018	161
CSPPSU-10vinylon	1.010	69	0.245	1.008	119
Nafion212	1.005	24	1.24	1.029	42

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
