# Peer review of "Crosslinked Sulfonated Polyphenylsulfone-Vinylon (CSPPSU-vinylon) Membranes for PEM Fuel Cells from SPPSU and Polyvinyl Alcohol (PVA)"

_polymers, 2020, doi:10.3390/polym12061354_

Round 1

Reviewer 1 Report

Kim and coauthors reported PVA-modified CSPPSU polymers with different PVA adding ratios. The chemical, mechanical, conductivity, and fuel cell properties of these polymer membranes were studied. The conductivity of the CSPPSU-10vinylon membrane is higher than that of the CSPPSU membrane. They use these polymers to fabricate fuel cells and find the CSPPSU-10vinylon-based one exhibits highest current density compared with CSPPSU and Nafion212 membranes. These findings are helpful to designing non-perfluorinated proton exchange membranes. I suggest the publication of this manuscript in Polymer with following issues to be revised.

  1. The chemical structure of these CSPPSU-nvinylon polymers should be shown.
  2. In figure 5(b), the CSPPSU-20vinylon shows lower conductivity than CSPPSU-5vinylon at 20% humidity. But why under improved humidity, the results are opposite?
  3. The chemical stability measurement results of these membranes should be presented.

Reviewer 2 Report

The manuscript by J.D. Kim et al. reports the fabrication and the characterization of proton-exchange membranes based on crosslinked composite obtained from sulfonated polyphenylsulfone and polyvinyl alcohol. The obtained polymer composites were extensively characterized by physicochemical methods. The ionic conductivities at temperatures higher than 100 oC were estimated. Finally, the performances of membranes were evaluated in PEMFC. The presence of vinylon led to the enhancement of water retention inside the membrane in an environment of low humidification. The excellent manuscript should be accepted in a present form.

I saw only one misprint: Sigaracet (page 4, line 145).
